# A Language Model with Limited Memory Capacity Captures Interference in Human Sentence Processing

**William Timkey**
New York University
wpt2011@nyu.edu

**Tal Linzen**
New York University
linzen@nyu.edu

## Abstract

Two of the central factors believed to underpin human sentence processing difficulty are expectations and retrieval from working memory. A recent attempt to create a unified cognitive model integrating these two factors relied on the parallels between the self-attention mechanism of transformer language models and *cue-based retrieval* theories of working memory in human sentence processing (Ryu and Lewis, 2021). While Ryu and Lewis show that attention patterns in specialized attention heads of GPT-2 are consistent with *similarity-based interference*, a key prediction of cue-based retrieval models, their method requires identifying syntactically specialized attention heads, and makes the cognitively implausible assumption that hundreds of memory retrieval operations take place in parallel. In the present work, we develop a recurrent neural language model with a single self-attention head, which more closely parallels the memory system assumed by cognitive theories. We show that our model's single attention head captures semantic and syntactic interference effects observed in human experiments.

## 1 Introduction

Theories of human sentence processing can be divided into two broad categories. *Expectation-based* theories like surprisal theory (Hale, 2001; Levy, 2008; Smith and Levy, 2013) ascribe processing difficulty to a word's predictability. *Memory-based* theories, such as cue-based retrieval (Van Dyke and Lewis, 2003; Lewis and Vasishth, 2005; Lewis et al., 2006; Wagers et al., 2009; Vasishth and Engelmann, 2021), account for processing difficulty incurred due to limitations on working memory encoding and retrieval. Integrating these approaches has become a major goal of sentence processing research (Demberg and Keller, 2009; Levy, 2013; Campanelli et al., 2018; Futrell et al., 2020; Hahn et al., 2022).

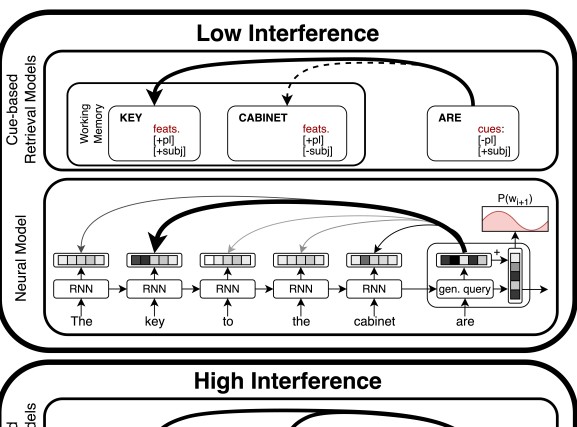
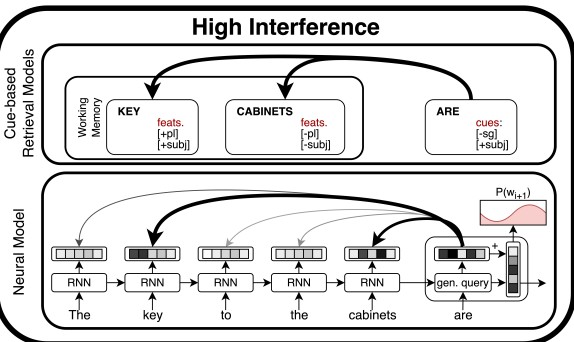

Figure 1: A demonstration of how the CBR-RNN can model similarity-based interference at the verb "are", compared to cue-based retrieval theories. While cue-based retrieval theories traditionally stipulate retrieval cue features, the neural model learns cue and target features through a next word and syntactic prediction objective. The model retrieves a weighted sum of past representations based on their similarity to the verb's *cue/query* feature vector. The retrieved representation is used to predict the next word in the sequence.

Surprisal theory argues that the processing difficulty associated with a word is linearly related to its negative log probability, or surprisal, in context. Surprisal theory is agnostic by design to the precise mechanisms underlying sentence processing; models with drastically different representational assumptions can be used to obtain surprisal estimates (Levy, 2008, 2013). The advent of neural network language models (NLMs), in particular, has broadened surprisal theory's empirical coverage to a wide range of phenomena (van Schijn-

del and Linzen, 2018; Arehalli and Linzen, 2020; Wilcox et al., 2021). However, NLMs' opaque internal representations make it difficult to relate them to theories of how language is represented and retrieved from memory by humans (though see Lakretz et al. 2019, 2021).

By contrast, cue-based retrieval theories make explicit claims about how working memory is structured and accessed during language comprehension. According to these theories, humans have a narrow attentional focus, and therefore must retrieve items from working memory to resolve long-distance linguistic dependencies. Items in working memory are accessed by matching a set of featural retrieval cues provided by the current word in parallel against the features of all items in working memory. The item which best matches the cue features is retrieved and used to parse the current word. The speed and accuracy of retrieval is affected by the number of items in memory that match the retrieval cue features. The effect of similarity of the target and distractor items on retrieval difficulty, referred to as *similarity-based interference*, is a key prediction of cue-based retrieval models, and has been used to explain various phenomena in the sentence processing literature (Vasishth and Engelmann, 2021).

Despite their empirical successes, implemented models of cue-based retrieval traditionally employ only a small set of hand-picked linguistic features, such as [+/-subject] or [+/-singular]. This limits model coverage, increases modeler degrees of freedom and decreases theoretical parsimony (Smith and Vasishth, 2020). By contrast, NLMs learn all of their linguistic features implicitly as they optimize a word prediction training objective.

Leveraging this ability of NLMs, Ryu and Lewis (2021) propose a synthesis of surprisal and cue-based retrieval theories. They note the parallels between cue-based retrieval and neural self-attention in transformer NLMs (Vaswani et al., 2017). To generate a contextualized representation of the current word, self-attention heads score the representations of all context words against a query generated at the current word, and sum the context word representations weighted by these attention scores. Transformer NLMs, such as GPT-2 (Radford et al., 2019), use many attention heads simultaneously, arranged into multiple layers. Ryu and Lewis show that attention patterns in a syntactically-specialized head in GPT-2 are consistent with similarity-based interference explanations of sentence processing difficulty. Crucially, in their model, all of the features used to compute attention scores were implicitly learned through the model's next word prediction training objective; no linguistic features need to be hand-picked.

Despite some parallels between transformers like GPT-2 and cue-based retrieval models, however, there are crucial differences in their working memory limitations. The smallest GPT-2 model has 144 attention heads. Under the view of self-attention as cue-based retrieval, this would mean that 144 distinct retrievals from working memory take place at each timestep. By contrast, cue-based retrieval theories argue that working memory is highly constrained; a very limited number of items can be retrieved from memory within the time it takes to process a single word. If an NLM is to truly integrate expectation and memory-based theories, then it must be consistent with assumptions about human working memory.

In this work, we present a recurrent NLM with memory limitations that are more closely aligned with cue-based retrieval theories. Figure 1 shows how we conceptualize memory retrieval in our model. We use self-attention to implement cue-based memory retrieval, but employ only a single attention head, limiting the model to a single retrieval operation per timestep. In contrast to prior approaches, we do not need to rely on post-hoc identification of syntactically specialized attention heads.

In our first experiment, we confirm that our model's single attention head learns to track subject-verb dependencies without direct supervision. In our main experiment, we show that measures of memory retrieval difficulty derived from our model's attention mechanism capture both semantic and syntactic retrieval interference effects from a human experiment (Laurinavichyute and von der Malsburg, 2022), without the need to hand-pick any linguistic features as in existing cue-based retrieval theories.[1]

## 2 Background: Attraction and Similarity-Based Interference

### 2.1 Agreement Attraction

In Mainstream English, the subject of a sentence must agree with its corresponding verb in num-

[1]All model and analysis code is available at: github.com/wtimkey/cue-based-retrieval-rnn

ber. For example, "The **keys are** on the table" is grammatical, but "The **key are** on the table" is ungrammatical. Although, grammatically speaking, number marking on the verb should depend only on the number of the subject noun phrase, real-time production experiments have shown that humans sometimes produce verb forms agreeing with a non-subject noun, called an *attractor*, instead of the true subject. For example, when prompted with a preamble such as "the key **to the cabinets**...", which contains a singular subject and a plural attractor noun, speakers often produce the plural verb "are" rather than the grammatically correct singular form "is" (Bock and Miller, 1991). This phenomenon is referred to as *agreement attraction*.

Similar phenomena have been observed in comprehension (Pearlmutter et al., 1999; Wagers et al., 2009). In general, readers slow down upon encountering a verb which does not agree in number with the subject, suggesting they are able to detect the agreement error. However, this slowdown is attenuated in the presence of an attractor, indicating the the attractor interferes with readers' ability to detect the agreement error. For example, the verb "are" is read faster in sentences like (2) than sentences like (1).

(1) The **key** to the cabinet are on the table.
(2) The **key** to the cabinets are on the table.

### 2.1.1 Similarity-based Interference

Agreement attraction in comprehension has been taken as evidence for cue-based retrieval theories of sentence processing (Wagers et al., 2009; Vasishth and Engelmann, 2021). These theories posit that humans have a sharply limited attentional focus, and must resolve syntactic dependencies by retrieving items from working memory. Retrieval is carried out though a parallel feature-matching process between items in memory and a set of cues for retrieval provided by the current word. The item which best matches the retrieval cues is retrieved and used to parse the current word.

Memory retrieval in cue-based theories is an imperfect process. The speed and accuracy of retrieval is affected by the number of items in memory that provide a partial match to the retrieval cues. This is referred to as *similarity-based interference*. If multiple items partially match the retrieval cues, then retrieval will be faster, but may be erroneous. For example, when a reader reaches the plural verb "are" in (2) and (1), they use the cues [+subject] and

[+plural] to retrieve the subject. In (1), only one item in memory, the "key" noun phrase, matches any of the retrieval cues (specifically, [+subject]). In (2), "key" matches one cue, [+subject], but "cabinets" matches the other cue, [+plural]. In this case, both items are possible candidates for retrieval, leading to a speedup in processing (Figure 1).

There are several explanations for this speedup (Wagers et al., 2009). We adopt the following explanation: In trials where the subject is correctly retrieved, the number-mismatch is detected, triggering a costly reanalysis process. This is the only expected outcome in (1). In (2), the non-subject, number-matching attractor can be mistakenly retrieved. In these cases the number agreement error goes unnoticed, and the reanalysis process is avoided.

### 2.2 Semantic Attraction

Attraction arises in the processing not only of syntactic constraints like subject-verb agreement, but also of semantic plausibility constraints (Cunnings and Sturt, 2018; Laurinavichyute and von der Malsburg, 2022). For example, consider the following pair of sentences:

(3) The **drawer** with the handle cut the bread.
(4) The **drawer** with the knife cut the bread.

Although both sentences involve a semantically implausible scenario in which a drawer is cutting bread, readers judge sentences like (4), which includes the attractor "knife", as more plausible than sentences like (3).

The cue responsible for agreement attraction is the same for any number-marked verb ([+/-plural]). Semantic retrieval cues, by contrast, can be lexically-specific. For example, the verb "cut" in (3) and (4) uses the semantic cue [+can_cut] when retrieving the subject (Laurinavichyute and von der Malsburg, 2022). In (4), the attractor "knife" matches the [+can_cut] cue, while the true subject "drawer" matches only the [+subject] cue, resulting in retrieval interference.

While it is possible that an array of highly specific features are used to access items in working memory, it is difficult to identify the right feature inventory in a principled way. Motivated by this problem, Smith and Vasishth (2020) showed that a cue-based retrieval model augmented with distributed semantic features from static word embeddings were able to capture semantic attraction

effects. In contrast to their model, NLMs with self-attention implicitly learn features to access past representations which are optimal for predicting upcoming words. In the present work, we evaluate whether such a model is sensitive to both semantic and syntactic interference.

## 3 Model

### 3.1 Motivation: Self-Attention as Cue-Based Retrieval

As Ryu and Lewis (2021) point out, self-attention (Luong et al., 2015; Vaswani et al., 2017) has key similarities to working memory in cue-based retrieval models: The query vector resembles the set of cue features in cue-based retrieval models in that both are matched against items in memory to determine what should be retrieved; the key vectors correspond to the featural representation of memory items in cue-based theories; and the attention score in self-attention and item activations in cue-based theories are both computed based on featural similarity.

There are key differences between cue-based retrieval theories and self-attention in transformers as well. The difference which motivates the present work is the number of retrieval operations that take place at each word. Cue-based retrieval theories of sentence processing are built upon independently-motivated principles of human memory. One principle is that humans can only keep one to three items in their attentional focus a time (McElree, 2001). In order to bring another item into focus, it must be retrieved from working memory, which is costly and imperfect. Minimizing the number of retrievals per word (to just one or two) is a guiding principle of the retrieval model proposed in (Lewis and Vasishth, 2005, p.412). The model Ryu and Lewis (2021) investigate, GPT-2 small, has 144 self-attention heads. Under the view that each head engages in cue-based retrieval, 144 distinct retrieval operations will take place at each word. In our single-headed model, exactly one retrieval takes place at each word.

In addition to cognitive plausibility considerations, it is unclear how a single measure of memory interference should be derived from a model that uses multiple specialized attention heads simultaneously. Ryu and Lewis (2021) identify and analyze a small subset of heads whose attention patterns correlated with subject-verb and reflexive dependencies. Some syntactic dependency types may

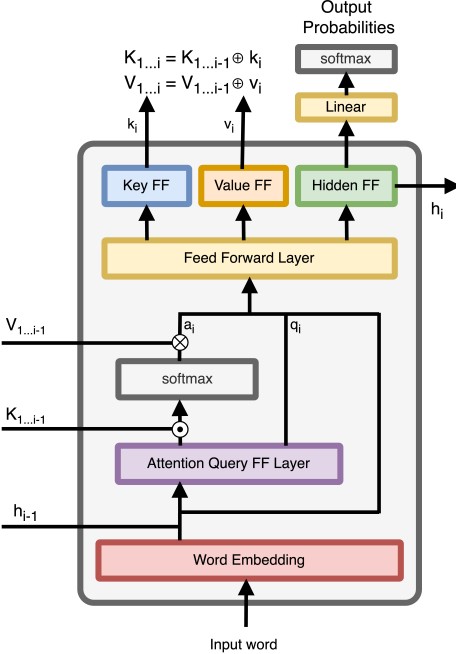

Figure 2: A schematic of the CBR-RNN Cell.

not be specialized to any particular head at all, but rather distributed across multiple interacting heads, obscuring interference effects.

Even if multiple attention heads correlate with some dependency type, each correlation could arise for different reasons. For example, we might find two heads whose attention patterns correlate with subject-verb dependencies in typical cases where the subject is semantically plausible and agrees in number with the verb. But one of these heads might identify the subject only by number marking, while the other uses only semantic information. One head would show only agreement interference, while the other shows only semantic interference. In human experiments, Laurinavichyute and von der Malsburg (2022) show that both types of interference interact with one another, producing an additive effect. It is not clear how the individual effects of each heads could be combined into a general measure of processing difficulty (though see Oh and Schuler 2022). Our single-headed model obviates the need to identify or aggregate attention heads.

### 3.2 Model Architecture

Our model, which we refer to as a Cue-Based Retrieval/Recurrent Neural Network (CBR-RNN), is a simple recurrent neural network (SRNN) language model (Elman, 1990) augmented with self-attention (Bahdanau et al., 2015). At each timestep, the model retrieves a weighted sum of representa-

tions computed in previous timesteps. The weights of this sum are determined through self-attention. The model is similar to Single Head Attention RNNs (SHA-RNNs; Merity 2019), with two differences motivated by cue-based retrieval theories.

First, in contrast with the SHA-RNN, which is based on long short-term memory (LSTM) units (Hochreiter and Schmidhuber, 1997), our model is based on an SRNN. LSTMs have memory cells designed explicitly to handle long-distance dependencies; by using the simpler SRNN architecture, which does not have this mechanism, we aim to limit our model's ability to encode information in its hidden state over long distances, forcing it to rely on self-attention. Second, in GPT-2 and SHA-RNNs, attention spans over representations from both the current and all previous timesteps. In our model, only key and value vectors from previous timesteps are accessed by the attention mechanism. This is because in cue-based retrieval theories, the current word is already active, so it does not need to be retrieved. This also allows us to generate key and value vectors for the current word that depend on the result of the current attention step. Likewise, in cue-based retrieval models, new items in working memory are not generated at the same time as retrieval cues, but after the retrieval operations have taken place.

Figure 2 provides a basic schematic of the model. At each timestep $i$, the embedding of the current word $w_i$ is concatenated with the previous hidden state, $h_{i-1}$. The result passes through a feedforward layer to produce an attention query vector $q_i$. Attention scores are generated by taking the dot product of $q_i$ and key vectors from all *previous* timesteps, $K_{1...i-1}$, then applying the softmax to produce attention weights. The result is the context vector $a_i$, an attention-weighted sum of value vectors $V_{1...i-1}$. Then, $g_i$, $w_i$, $q_i$ and $v_t$ are concatenated and passed through two feedforward layers. The resulting vector is split into three smaller vectors of equal size: the key and value vectors for the current timestep ($k_i$ and $v_i$, respectively) and the hidden representation $h_i$. We append $k_i$ and $v_i$ to the key and value caches, $K$ and $V$, for use in future timesteps. We use $h_i$ for all predictions, and pass it to the following timestep.

### 3.3 Model Training

We trained our model on two objectives. The first is next-word prediction on a lowercased version of the 103M token WikiText-103 corpus (Merity et al., 2017). In contrast to many contemporary language models that use subword tokenization, we use a simpler word-level tokenization scheme, with a vocabulary pruned to the most frequent 50k word types from the training corpus.

The second objective is Combinatory Categorical Grammar (CCG) supertagging (Steedman, 1987). Prior work has argued that humans weigh syntactic factors more heavily than NLMs do when making predictions (van Schijndel and Linzen, 2021; Arehalli et al., 2022). Since we are interested in modeling how humans use working memory when processing syntactic dependencies, we aim to encourage our model to represent and use syntactic information its attention mechanism. The addition of a CCG supertagging objective has been shown in prior work to induce richer syntactic representations (Enguehard et al., 2017). We generated CCG supertags for the entire WikiText-103 corpus using a state-of-the-art CCG supertagger (Tian et al., 2020).[2] The global loss, $L$, is defined as:

$$L = L_{LM} + \alpha L_{CCG} \qquad (1)$$

where $L_{LM}$ is the language modeling loss, $L_{CCG}$ is the CCG supertagging loss, and $\alpha$ is a scaling factor. Of the 15 total random seeds of our model, five were trained with $\alpha = 5$, five were trained with $\alpha = 1$, and five random seeds were trained with next-word prediction loss only ($\alpha = 0$).

### 3.4 Language Modeling and CCG Supertagging Performance

To test how well our model performs on its training objectives relative to existing architectures, we trained LSTMs with a similar number of parameters on the same data and objectives as the CBR-RNN models. We found that CBR-RNN models have generally comparable, or even superior language modeling and CCG supertagging performance to the LSTMs.

To test whether the self-attention mechanism is essential for the model's language modeling and CCG supertagging performance, we trained 15 CBR-RNN models with an ablated attention mechanism. The language modeling perplexity was significantly higher in the ablated models. By contrast, CCG supertagging accuracy did not significantly

---

[2]The accuracy of this tagger on the CCGBank corpus (Hockenmaier and Steedman, 2007) is 96.1%.

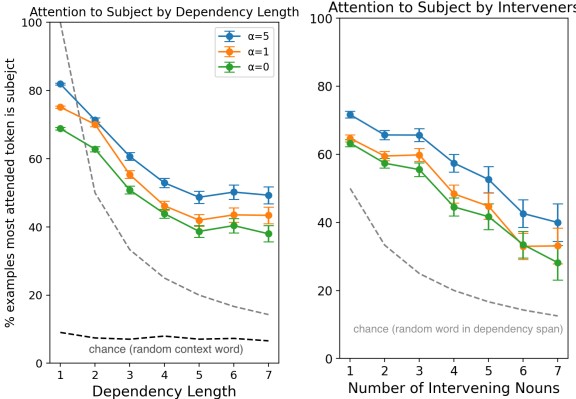

Figure 3: Percentage of items in the subject-verb dependency corpus in which the subject representation has the largest attention weight upon encountering the verb, plotted by dependency length (left) and number of intervening nouns (right). Chance is given by dashed lines. The darker dashed line is chance when picking any left-context token at random, the lighter dashed line is picking any noun within the span of the dependency at random. $\alpha$ is the weight of the CCG supertagging objective.

decrease in the ablated models. Detailed results for both experiments are reported in Appendix B.

## 4 Experiments

We first examined whether the attention mechanism in trained CBR-RNN models is sensitive to long-distance subject-verb dependencies, even though it does not receive direct supervision in training as to the words that should be attended to. Then, in our main experiment, we evaluated whether language model surprisal and a measure of similarity-based interference from our model's attention mechanism capture semantic and agreement attraction effects using experimental stimuli from Laurinavichyute and von der Malsburg (2022).

### 4.1 Does Self-Attention Track Subject-Verb Dependencies?

Cue based retrieval models of sentence processing assume that items are retrieved from working memory to guide parsing and to form syntactic dependencies. In this experiment, we verify that the behavior of our models attention mechanism is generally consistent with this this goal. We investigate whether attention patterns from our model reflect a sensitivity to subject-verb dependencies of varying lengths and number of intervening nouns.

We generated a dependency corpus from the test and validation portions of our training corpus us-

ing the spaCy dependency parser from the spaCy `en_core_web_trf` pipeline [3]. We extracted all examples of subject-verb dependencies (approximately 17k in total) along with their length and the number of nouns that intervened between the subject and the verb.

We evaluate the percentage of examples where the models assigns a greater attention weight to the subject than to any other words upon encountering the verb. We report this percentage as a function of both dependency length and the number of intervening nouns between the verb and its subject. We compare these percentages to two baselines: randomly selecting any context token, and randomly selecting any noun within the dependency span.

#### 4.1.1 Results

Models pay significantly more attention to the current verb's subject than chance would predict (Figure 3). Models trained with the additional CCG supertagging objective attended to the subject more often than models trained without it. This suggests that the auxiliary syntactic objective is useful for inducing syntax-sensitive behavior in the model's attention mechanism.

### 4.2 Do Models Capture Agreement Attraction and Semantic Attraction?

In the following section, we evaluate whether our model captures agreement and semantic attraction, and their interaction with one another. We examine surprisal and a measure of similarity-based interference from our model's attention head. We also evaluate the GPT-2 model used by Ryu and Lewis on both measures, using the syntactically-specialized heads the authors identify, referred to by their location as `head4_3`, `head3_6`, `head6_0`, and `head2_9`.

#### 4.2.1 Materials

We evaluate our models on experimental stimuli from Laurinavichyute and von der Malsburg (2022) Experiment 3. We made some minor modifications to the stimuli to eliminate potential confounds specific to our model (seeAppendix A for details). Examples of each condition are given in Table 1. All conditions violate either subject-verb agreement (A, B), semantic plausibility (C, D) or both constraints simultaneously (E–H). Each violation type has a baseline condition with no attractor, and

---

[3]The accuracy of this parser on the Penn Treebank is 95.1%.

| Condition | Violation | Attractor | Prefix | Verb |
|-----------|-----------|-----------|--------|------|
| A. | Agreement | None | The drawer with the handle really | OPEN |
| B. | Agreement | Agreement | The drawer with the handles really | OPEN |
| C. | Semantic | None | The drawer with the handle really | CUTS |
| D. | Semantic | Semantic | The drawer with the knife really | CUTS |
| E. | Double | None | The drawer with the handle really | CUT |
| F. | Double | Double | The drawer with the knives really | CUT |
| G. | Double | Semantic | The drawer with the knife really | CUT |
| H. | Double | Agreement | The drawer with the handles really | CUT |

Table 1: Example stimuli from Experiment 3 of Laurinavichyute and von der Malsburg (2022).

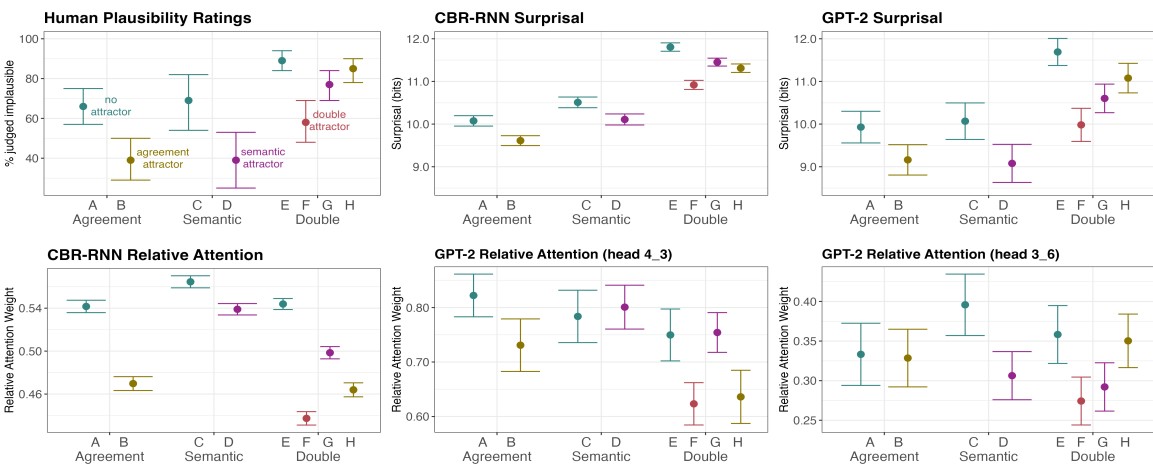

Figure 4: Simulation results for Experiment 3 of Laurinavichyute and von der Malsburg (2022). Human results (top left) are the percentage of participants who (correctly) ruled the verb as an implausible continuation. Error bars indicate 95% confidence intervals.

a condition with an attractor which matches the verb in the violated feature(s). For example, in the double-violation, no-attraction condition (D), both the subject "drawer" and the attractor "handle" are semantically implausible and do not match the number marking of the verb "cut". In the double-violation, semantic-attraction condition (G), the attractor "knife" is a semantically plausible subject but does not match the number marking of the verb. In the double-violation, double-attraction condition (F), the attractor "knives" is both semantically plausible and matches the verb's number marking.

Participants in Laurinavichyute and von der Malsburg's human study were asked to first memorize the verb before seeing a preamble. They were then asked to rate the plausibility of the verb as a continuation of the preamble. If items with an agreement attractor (B) are rated more plausible than items without an agreement attractor (A), then this is taken as evidence for agreement attraction. If items with a semantic attractor (D) are rated more plausible than items without a semantic attractor (C), then this is evidence for semantic attraction. If

items with a simultaneous agreement and semantic attractor (F) are rated more plausible than items with a semantic or agreement-only attractor (G,H), this is evidence that that semantic and agreement attraction can occur simultaneously.

Laurinavichyute and von der Malsburg found that participants were more likely to rate the continuation as plausible if a distractor agreed in a semantic or number feature with the verb. Additionally, the effects of agreement and semantic attraction were generally additive in the case of double violations. This provides evidence both for the existence of semantic attraction effects, and for cue-based retrieval models more generally, which can account for both types of attraction and their additive effects if semantic and agreement features are used simultaneously as retrieval cues.

#### 4.2.2 Linking Hypothesis for Plausibility Ratings

At each word, our model calculates attention weights between the current word and all past representations. This is a measure of the relative strength

of association between the current word's query vector and past representations in working memory. These weights sum to 1, and can be thought of as modeling the distribution of retrieval probabilities for items in working memory.

If the true subject has a high retrieval probability, then it will be retrieved often, and participants will be more likely to notice the semantic plausibility or agreement violation. The higher the retrieval probability of the non-subject noun, the more likely the violation is to go unnoticed. We model the probability of a human judging the sentence as implausible as the attention weight assigned to the subject, normalized by the total attention weight of the subject and non-subject nouns:

$$RelAttn(v, s) = \frac{Attn(v, s)}{Attn(v, s) + Attn(v, n)} \quad (2)$$

Where $Attn(a, b)$ denotes the attention weight of $b$ in memory when word $a$ is the query. The subject is denoted as $s$, $v$ denotes the subject's verb, and $n$ denotes the non-subject noun. This linking hypothesis differs from that of Laurinavichyute and von der Malsburg (2022), which relates rating a sentence as implausible to retrieval failure, in which no item in memory receives sufficient activation above some threshold. The retrieval threshold and activation noise are hyperparameters that must be fit to the experimental data. We chose our linking hypotheses because it is simpler, requiring no hyperparameter fitting.

We also report surprisal estimates from the CBR-RNN models. Our linking hypothesis between surprisal and plausibility ratings is simply that more surprising continuations should be judged implausible more often than less surprising ones. We see the goal of the present work as capturing the qualitative direction of the effect, and leave the construction of linking hypotheses with precise quantitative predictions of processing difficulty for future work.

### 4.2.3 Results

Results are summarized graphically in Figure 4. All significance tests were conducted using linear mixed-effects models with random intercepts for model instances and experimental items, and are reported in Appendix C. The qualitative pattern of attraction effects did not differ across the three CCG supertagging loss weights ($\alpha = 0, \alpha = 1, \alpha = 5$). Therefore, we report results from all models together. Results from each training condition are reported graphically in Appendix D.

**Single Violation:** Relative attention and surprisal were significantly lower in both the agreement (B) and semantic attractor (D) conditions than in the no attraction conditions (A) and (C). Consistent with the human pattern, this demonstrates both semantic and syntactic attraction. In the human experiment, there was no difference in the size of the attraction effect between the agreement and semantic conditions. Surprisal from our model replicated this finding but relative attention did not; when computed using relative attention, the agreement attraction effect was stronger than the semantic attraction effect.

**Double Violation:** We found a significant attraction effect in each of the single-feature attraction conditions in both relative target attention and surprisal, consistent with the human data. There was no significant difference in the size of agreement and semantic attraction effects in the human data. This was replicated in the surprisal estimates, but was not replicated in relative attention. There was no significant interaction between semantic and agreement attraction (E-H = G-F) in either surprisal or relative attention. In other words, the effect of attraction was additive, which is consistent with the human pattern.

**GPT-2 Results:** In both the single and double violation conditions, the GPT-2 surprisal pattern was consistent both with the human data and surprisal from our model. However, in relative attention, one attention head, `head4_3` exhibited an agreement attraction effect, but no semantic attraction effect, while another attention head, `head3_6` showed the opposite pattern. These results show that different attention heads can exhibit complementary interference effects for a single dependency type, highlighting the practical difficulty of deriving a single attention-based measure of interference when a model has multiple attention heads. Results from the other two heads are reported in Appendix C.

## 5 Discussion

### 5.1 Discrepancies Between Model Predictions and Human Behavior

While relative attention correctly predicted the direction of both semantic and syntactic attraction in subject-verb dependencies, there were a few quantitative discrepancies between the model and human results. First, human plausibility ratings in the "no attractor" conditions were higher across

the board (75–90%) than relative attention would predict (0.53–0.56). Even when there was no attractor present, the attention weight of the subject remained slightly above 50%. An inspection of the attention weights showed that the models often attend broadly to many past memory representations, rather than only attending to the subject and attractor nouns. Our models retrieve a weighted sum over past representations, rather than a single item. Attending broadly may be the most useful strategy for the prediction objective, and there is no constraint preventing models from doing so. This tendency may be counteracted by incorporating an attention sparsity regularization term in training (Zhang et al., 2019). This type of regularization can be thought of as encouraging the model to be more certain about which past representation will be most useful for next word prediction.

Relative attention from our model also predicts a smaller effect for semantic attraction than agreement attraction, while no difference was observed in the human study. This discrepancy appeared in models trained with and without the syntactic auxiliary objective. One possibility is that semantic attraction effects are better explained by expectation violation than retrieval interference. Surprisal measures more closely matched the semantic attraction patterns. Evidence from agreement attraction studies suggests that memory retrieval may be initiated as part of an error-driven repair process when expectations are violated (Wagers et al., 2009; Schlueter et al., 2019). Campanelli et al. (2018) also show that retrieval interference is modulated by the predictability of the word at the retrieval site. Future work should explore how measures of retrieval difficulty might be more explicitly augmented by predictability.

### 5.2 Relationship to Resource-Rational Models

Our model can be related to resource rational theories of cognition (Lieder and Griffiths, 2020), and especially the model of sentence processing proposed by Hahn et al. (2022), who propose a unification of expectation and memory-based theories under the hypothesis that limited memory resources are optimally allocated to maximize the predictability of future material. Consistent with this account, our model has the computational goal of maximizing predictability. The model accomplishes this goal in part by utilizing its memory retrieval mechanism. Given that retrieval capacity is limited, the

model must learn to retrieve items from memory which are optimal for prediction, and will learn a set of latent features which enable this optimal retrieval. For example, when the model encounters a verb, its subject may be the best item to retrieve in order to predict the verb's continuation. To reliably retrieve subjects, the model would need to learn a latent [+/-subject] feature.

The account proposed by Hahn et al. (2022) is a *computational level* theory (Marr, 1982), meaning it specifies the computational problem the human parser is solving, but is agnostic to how the theory is implemented in a model or the mind. A central motivation of the present work is to combine the computational goal outlined by resource-rational models (updating one's beliefs about a sentence by making rational use of finite memory resources), with a well-established, independently motivated *algorithmic level* level theory of working memory access.

## 6  Conclusion

In this work, we proposed a neural model which incorporates memory constraints from cue-based retrieval theories. We showed that the model's attention mechanism tracks subject-verb dependencies over long distances and with intervening distractor nouns. Finally, we evaluated whether the model's attention component tracks similarity-based interference effects found in human attraction experiments. We showed that both model surprisal estimates and relative attention from our model's self-attention mechanism both predicted the presence of semantic and agreement attraction, without the need to explicitly stipulate any linguistic features in the model. However, relative attention generally under-predicted plausibility ratings and the relative size of the semantic attraction effect. Taken together, we showed that mechanistic explanations of processing difficulty rooted in memory retrieval constraints can emerge as a by-product of minimizing the surprisal of upcoming words.

## Ethics Statement

The authors foresee no ethical concerns with the present work.

## Acknowledgements

This material is based upon work supported by the National Science Foundation (NSF) under Grant No. BCS-2020945, and in part through the NYU IT

High Performance Computing resources, services, and staff expertise. We would also like to thank Suhas Arehalli and members of the NYU Computation and Psycholinguistics lab for their insightful feedback and thought-provoking discussion.

## Limitations

### 6.1 Dependency Types

In this work, we investigated interference effects in subject-verb dependencies. We chose this dependency type because it is one of the most widely studied, but it is far from the only dependency type. In future work, we aim to evaluate our model on various dependency types, such as reflexive anaphora (Dillon et al., 2013), and filler-gap dependencies (McElree, 2001).

### 6.2 Assumptions About Working Memory

While our model address a major assumption of cue-based retrieval models, limited retrieval capacity, there remain several key differences between the two. We describe some of these differences and how they may be addressed in this section.

Cue-based retrieval theories typically assume that items stored in working memory are individual syntactic constituents (Lewis and Vasishth, 2005; Lewis et al., 2006; Dotlačil, 2021), while our model assumes items in working memory are representations of previous timesteps. One potential direction for future work is forcing models to explicitly represent syntactic structure in short-term memory. This could be achieved through a syntactic attention masking mechanism similar to that of Sartran et al. (2022).

Some cue-based retrieval models assume that items in working memory decay over time, making retrieval of older material more difficult than newer material. Decay effects can explain findings like Van Dyke and Lewis (2003), in which the difficulty of ambiguity resolution in garden path sentences is modulated by the length of the ambiguous region. Recent work has also found that limiting context length in neural language models improves their fit to human reading data (Kuribayashi et al., 2022). In future work, graded memory decay could be incorporated into the model by applying a soft attention mask to older material.

### 6.3 Interpretable Linguistic Features

While the CBR-RNN may capture similarity-based interference effects, it is not clear if it accomplishes this by implicitly learning features corresponding to those posited in existing cue-based retrieval models like [+/-subject] or [+/-animate]. Causal representational probing methods may prove useful for determining whether our models implicitly represent and use features which are consistent with existing theories (Elazar et al., 2021; Ravfogel et al., 2021).

### 6.4 Training Data Limitations

While we train our models on a developmentally-plausible amount of text, our training corpus consists entirely of articles from Wikipedia, which is quite different from the data children receive when acquiring language. Mueller and Linzen (2023) show that training language models on child-directed speech leads to more rapid humanlike generalization. Future work could investigate whether training CBR-RNNs on more developmentally-plausible corpora leads to more humanlike memory retrieval behaviors.

Unlike humans, our model learn from text alone. Lexical features implicated in semantic attraction experiments, like [+/-can_cut] or [+/-shatterable] are likely learned by humans with the help of visual information and interaction with the world, sources of information which are not available to our model.

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

| Original | Replacement |
|---|---|
| landscape(s) | mountain(s) |
| highrise | apartment |
| crackle | warm |
| loft(s) | balcony/ie(s) |
| dent(s) | scratch(es) |
| glow(s) | shine(s) |
| dribble | drip |
| soothingly | comfortably |

Table 2: Replacement of out-of-vocabulary tokens with synonyms.

| Original | Replacement |
|---|---|
| tram stop | stop |
| wall calendar | calendar |
| chocolate fountain | fountain |
| winter garden | garden |
| coffee shop | cafe |
| towel hook | mirror |
| light switch | switch |
| license plate | plate |
| walkie talkie(s) | radio(s) |

Table 3: Replacement of noun-noun compounds with single nouns.

## A Modification to Stimuli

We made some minor modifications to the materials of Laurinavichyute and von der Malsburg (2022) to make them compatible with our model. Firstly, some words in the stimuli were outside of the model's vocabulary. We replaced all such words with synonyms. Additionally, we replaced all noun-noun compounds with either a single-word synonym just the head noun of the compound. This was because our model often distributed attention across both nouns in the compound, artificially deflating the level of attraction in certain conditions. Synonym replacements are reported in Table 2, noun-noun compound replacements are reported in Table 3.

## B Perplexity and CCG Supertagging Performance

Baseline models are matched to the three syntactic conditions ($\alpha = 0$, $\alpha = 1$, $\alpha = 5$). For the first baseline, we ablate the attention mechanism from the CBR-RNN during training. For the second baseline, we train a single-layer LSTM (with no attention) whose hidden dimensionality (256) matches that of the CBR-RNN models. For the third baseline, we train a two-layer LSTM (with no attention) whose parameter count (14.9M) approximately matches that of the CBR-RNN.

We evaluated the perplexity of all models on the test portion of WikiText-103. Matched for auxiliary objective weight, CBR-RNN models all achieved lower perplexity than both the parameter-matched and hidden dimensionality-matched LSTM models (Table 4).

CCG supertagging accuracy was evaluated on the test portion of the CCGBank dataset. All models achieve relatively high accuracy in CCG tagging, with similar accuracies across all conditions. CBR-RNN models have generally comparable, or even superior language modeling and CCG supertagging performance to parameter-matched LSTM language models (Table 5).

## C Statistical Analysis

We conduct significance tests for agreement and semantic attraction effects and their interaction in each violation conditions, using both of our models measures, surprisal and relative attention. We use linear mixed effects models from the lmer package in R. with random intercepts for items and model

| Supertagging objective weight: | $\alpha = 0$ | $\alpha = 1$ | $\alpha = 5$ |
|---|---|---|---|
| **CBR-RNN ($d = 256$)** | 61.8 $\pm$0.4 | 63.3 $\pm$0.3 | 75.1 $\pm$0.5 |
| CBR-RNN ($d = 256$, no attention) | 75.1 $\pm$1.1 | 76.2 $\pm$1.2 | 92.3 $\pm$1.1 |
| LSTM ($d = 256$, single-layer) | 66.0 $\pm$2.7 | 67.6 $\pm$1.3 | 79.4 $\pm$0.6 |
| LSTM ($d = 271$, two-layer) | 61.2 $\pm$1.3 | 65.3 $\pm$7.1 | 77.6 $\pm$3.0 |

Table 4: Perplexity (with standard deviations) on the Wikitext-103 test set averaged across random seeds. Rows 2–4 are the three sets of baseline models.

| Supertagging obj. weight: | $\alpha = 1$ | $\alpha = 5$ |
|---|---|---|
| **CBR-RNN (d = 256)** | 84.4 $\pm$0.5 | 86.0 $\pm$0.1 |
| CBR-RNN (d=256, no attention) | 83.9 $\pm$1.5 | 85.1 $\pm$1.2 |
| LSTM (d=256,l=1) | 84.5 $\pm$0.7 | 85.7 $\pm$0.9 |
| LSTM (d=271,l=2) | 85.1 $\pm$3.3 | 86.2 $\pm$2.6 |

Table 5: CCG supertagging accuracies (with standard deviations) on the test position of CCGBank averaged across random seeds. The first row of results are the models under investigation in this study, rows 2-4 are the three sets of baseline models.

| Single Violation (A, B, C, D) | Effect | Surprisal | Relative Target Attention |
|---|---|---|---|
| Agreement Attraction | A - B | $\beta = -0.46, p < 0.001$ ✓ | $\beta = -0.07, p < 0.001$ ✓ |
| Semantic Attraction | C - D | $\beta = -0.40, p < 0.001$ ✓ | $\beta = +0.03, p < 0.001$ ✓ |
| Interaction of Attraction Type | (A - B) - (C - D) | $\beta = +0.06, p = 0.721$ ✓ | $\beta = +0.04, p < 0.001$ ✗ |
| Double Violation (E, F, G, H) | | | |
| Double Attraction | E - F | $\beta = -0.88, p < 0.001$ ✓ | $\beta = -0.11, p < 0.001$ ✓ |
| Agreement Attraction | E - H | $\beta = -0.49, p < 0.001$ ✓ | $\beta = -0.08, p < 0.001$ ✓ |
| Semantic Attraction | E - G | $\beta = -0.35, p < 0.001$ ✓ | $\beta = +0.05, p < 0.001$ ✓ |
| Interaction | (E - G) - (H - F) | $\beta = -0.03, p = 0.645$ ✓ | $\beta = +0.02, p = 0.053$ ✓ |

Table 6: Linear mixed effects modeling results for all combined CBR-RNN models ($\alpha = 0$, $\alpha = 1$, $\alpha = 5$) on Experiment 3 of Laurinavichyute and von der Malsburg (2022). Results marked with a ✓ are consistent with the human results, while results marked with a ✗ are inconsistent with the human results.

seeds. Results from linear mixed effects modeling of CBR-RNN results are reported in Table 6. Results from GPT-2 are reported in Table 7. Results from additional GPT-2 heads are visualized in Figure 6.

## D  Attraction Results by Syntactic Objective Weight

In Figure 5, we visualize results from the attraction experiments split across the three CCG supertagging training conditions ($alpha = 0$, $alpha = 1$, $alpha = 5$).

| Single Violation (A, B, C, D) | Effect | Surprisal | Relative Attention (head4_3) | (head3_6) |
|---|---|---|---|---|
| Agreement Attraction | A - B | $\beta = -0.77, p < 0.001$ ✓ | $\beta = -0.09, p < 0.001$ ✓ | $\beta = +0.00, p = 0.619$ ✗ |
| Semantic Attraction | C - D | $\beta = -0.99, p < 0.001$ ✓ | $\beta = +0.02, p = 0.672$ ✗ | $\beta = -0.09, p = 0.018$ ✓ |
| Interaction of Attraction Type | (A - B) - (C - D) | $\beta = -0.22, p = 0.729$ ✓ | $\beta = +0.11, p = 0.076$ ✓ | $\beta = +0.08, p = 0.069$ ✓ |
| **Double Violation (E, F, G, H)** | | | | |
| Double Attraction | E - F | $\beta = -1.70, p < 0.001$ ✓ | $\beta = -0.13, p = 0.024$ ✓ | $\beta = -0.08, p = 0.015$ ✓ |
| Agreement Attraction | E - H | $\beta = -0.61, p < 0.001$ ✓ | $\beta = -0.11, p < 0.001$ ✓ | $\beta = -0.01, p = 0.577$ ✗ |
| Semantic Attraction | E - G | $\beta = -1.08, p < 0.001$ ✓ | $\beta = +0.00, p = 0.914$ ✗ | $\beta = -0.06, p = 0.072$ ✗ |
| Interaction | (E - G) - (H - F) | $\beta = -0.01, p = 0.977$ ✓ | $\beta = -0.02, p = 0.787$ ✓ | $\beta = -0.01, p = 0.805$ ✓ |

| Single Violation (A, B, C, D) | Effect | (head6_0) | (head2_9) |
|---|---|---|---|
| Agreement Attraction | A - B | $\beta = -0.13, p < 0.001$ ✓ | $\beta = -0.01, p = 0.034$ ✓ |
| Semantic Attraction | C - D | $\beta = +0.01, p = 0.870$ ✗ | $\beta = +0.00, p = 0.743$ ✗ |
| Interaction of Attraction Type | (A - B) - (C - D) | $\beta = +0.14, p = 0.018$ ✗ | $\beta = +0.00, p = 0.787$ ✓ |
| **Double Violation (E, F, G, H)** | | | |
| Double Attraction | E - F | $\beta = -0.14, p = 0.005$ ✓ | $\beta = -0.03, p = 0.015$ ✓ |
| Agreement Attraction | E - H | $\beta = -0.11, p < 0.001$ ✓ | $\beta = -0.02, p < 0.001$ ✓ |
| Semantic Attraction | E - G | $\beta = -0.01, p = 0.856$ ✓ | $\beta = -0.01, p = 0.668$ ✗ |
| Interaction | (E - G) - (H - F) | $\beta = -0.03, p = 0.516$ ✓ | $\beta = -0.00, p = 0.603$ ✓ |

Table 7: Linear mixed effects modeling results for surprisal and relative attention (by head) from GPT-2 on Experiment 3 of Laurinavichyute and von der Malsburg (2022). Results marked with a ✓ are consistent with the human results, while results marked with a ✗ are inconsistent with the human results.

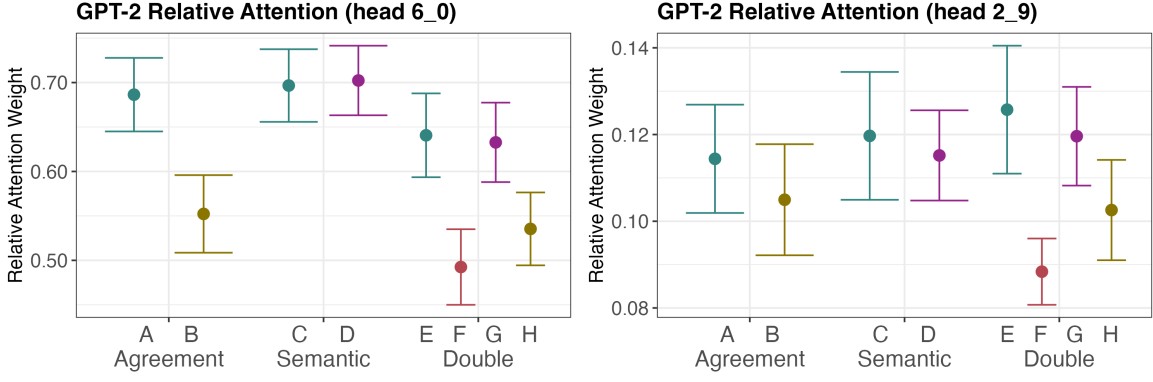

Figure 5: Surprisal and relative attention results in each of the three CCG supertagging conditions.

Figure 6: Surprisal and relative attention results from two additional GPT-2 heads.