# OpenReview forum: "A Language Model with Limited Memory Capacity Captures Interference in Human Sentence Processing"
_EMNLP/2023/Conference — EMNLP 2023 Findings_

### Official Review · Reviewer_jZo2 · 2023-08-03

**Soundness:** 3

**Excitement:**

3: Ambivalent: It has merits (e.g., it reports state-of-the-art results, the idea is nice), but there are key weaknesses (e.g., it describes incremental work), and it can significantly benefit from another round of revision. However, I won't object to accepting it if my co-reviewers champion it.

**Missing References:**

The reference list is sufficiently valid

**Paper Topic And Main Contributions:**

The paper proposes a recurrent neural network model augmented with a self-attention module for human-like sentence processing based on both expectation- and cue-based serial processing.
Results support the author(s)’ claim, although the whole process seems to be far from being cognitively plausible.

**Reasons To Accept:**

One of the major contribution of the paper is to rely on both expectation- and cue-based serial processing.
These two features have often been considered as either principle inspiring model feature.
Interestingly, the author(s) show(s) that both surprisal and memory interference help in predicting semantic or grammatical agreement violations in automatic sentence processing.

**Reasons To Reject:**

The paper is not thoroughly well written: it contains typos, and the introduction and related work sections are very long.
A more focussed reorganisation of the structure would be of great help for the reader.

**Reproducibility:**

2: Would be hard pressed to reproduce the results. The contribution depends on data that are simply not available outside the author's institution or consortium; not enough details are provided.

**Reviewer Confidence:**

3: Pretty sure, but there's a chance I missed something. Although I have a good feel for this area in general, I did not carefully check the paper's details, e.g., the math, experimental design, or novelty.

**Typos Grammar Style And Presentation Improvements:**

Abstract: an cognitively This should be “a cognitively”
Line 124: post-doc ?? Did you mean post-hoc?
Line 138: Mainstream English. I have never heard about an English variant where number agreement is not an issue. Moreover, subject-verb agreement is common of many many languages.

Line 29: Expectation-based theories like surprisal. This is a bit odd since it sounds as “surprisal” may be the name of the theory. Surprisal is  a quantitative measure for evaluating the complexity/difficulty of serial processing based on accruing expectations.

Figure 1 is not cited in the text.

---

> ### Author Rebuttal · Authors · 2023-08-28
>
> *“...[the paper] contains typos...”*
> * Thank you for identifying some typos and a missing figure reference in the paper. We will correct these in the final paper.
> ---
> *“...the introduction and related work sections are very long...”*
> * We feel that the introduction, which runs just over one page in total, is standard for EMNLP submissions. The paper does not contain a related work section; related work was incorporated into the introduction and background sections. In the camera-ready version of the paper, we will make the background more concise. However, because the paper is primarily concerned with connecting cognitive theories to computational modeling, some amount of theoretical discussion is unavoidable.
> ---
> *“...I have never heard about an English variant where number agreement is not an issue... subject-verb agreement is common of many many languages...”*
> * There are several varieties of English which have different agreement patterns than Mainstream English. One example is varieties of Northern British English with the Northern Subject Rule. In these varieties, both singular and plural non-pronominal subjects may take the -s verbal ending [1]. Another example is varieties of African American Vernacular English, in which there is no subject verb agreement, with the exception of the copula [2]. We will make this point clearer in the camera-ready version.
> * It is true that many languages have subject-verb agreement. While we only focused on agreement in Mainstream English in the paper, we will adjust the wording of line 138 to reflect the fact that agreement is pervasive across the world’s languages.
> ---
> *“...it sounds as 'surprisal' may be the name of the theory...”*
> * Surprisal theory is indeed a theory which characterizes the relationship between the *measure* surprisal (the negative log probability of an event) and difficulty in sentence processing. The theory has been formalized and theoretically motivated in several ways, such as the cost of resource-reallocation in a fully-parallel parser [3]. We regret any confusion that may have arisen from our mention of surprisal in line 28. To avoid future confusion, we will replace “surprisal” in line 28 with “surprisal theory” in the final paper.
> ---
> [1] Pietsch, L. 2005. *Variable Grammars: Verbal Agreement in Northern Dialects of English.* Berlin, Boston: Max Niemeyer Verlag. https://www.degruyter.com/document/doi/10.1515/9783110944556/html
>
> [2] Labov, W. 1998. *Coexistent systems in African-American vernacular English.* In Mufwene, S. S., Rickford, J. R., Bailey, G., & Baugh, J., editors, African-American English: Structure, history, and use, pages 110–153. Routledge, London, 1998.
>
> [3] Roger Levy. 2008. *Expectation-based syntactic comprehension.* Cognition, 106(3):1126–1177.

---

### Official Review · Reviewer_Cvhi · 2023-08-04

**Typos Grammar Style And Presentation Improvements:** From which model were the results in …
**Soundness:** 3

**Excitement:**

4: Strong: This paper deepens the understanding of some phenomenon or lowers the barriers to an existing research direction.

**Paper Topic And Main Contributions:**

Aiming for a cognitively plausible model of human sentence processing, this study introduces a model named CBRNN, which integrates forward-looking expectation and backward-looking memory constraints. Then, its human-like behaviors against context interference are tested on controlled sentences involving subject-verb agreement. The experiments show that (i) the CBRNN model trained with syntactic auxiliary objective assigns larger attention to a proper subject when processing the verb and (ii) the CBRNN model's attention and surprisal generally mirrors the human processing difficulty of the controlled sentences.

**Questions For The Authors:**

Will the model/data be publicly available?
At first glance, the relative attention results in Figure 4 do not mirror the human data; for example, humans can better judge the implausibility of verbs in the "double" condition, but relative attention does not. Why not focus on that part in Section 4.3.3?

**Reasons To Accept:**

- The motivation for model design and experiments is clear. Integrating expectation and memory retrieval in cognitive modeling is generally an important topic.
- The experimental design is well-controlled and adequate for evaluating the cue-based retrieval model.
- The results seem positive; that is, the proposed model can (partially) replicate the human sentence processing load.

**Reasons To Reject:**

- There is no comparison with baseline, e.g., multi-head attention Transformers, in main experiments; I'd like to know if the models' (partially) human-like behaviors (e.g., results in Figure 4) stem from the intended, special characteristics of CBRNN, such as RNN with memory-limited attention or not.
- Attention does not always reflect faithful attention. For example, if a particular value vector is close to zero, the corresponding attention is canceled [Kobayashi+,21]. Observing the attention weight would be a good starting point, but I'm afraid that the experiments missed some important aspects of the model's context retrieval beyond attention weight.

**Reproducibility:**

4: Could mostly reproduce the results, but there may be some variation because of sample variance or minor variations in their interpretation of the protocol or method.

**Reviewer Confidence:**

4: Quite sure. I tried to check the important points carefully. It's unlikely, though conceivable, that I missed something that should affect my ratings.

---

> ### Author Rebuttal · Authors · 2023-08-28
>
> *Comparison to a transformer baseline*
> * Thank you for the great suggestion to compare results from the CBRNN model to a transformer model. We will include comparisons to attention and surprisal measures from a transformer model in the final paper. However, we believe that results from multi-headed models would come with a couple of caveats, which we include both here and in response to another reviewer with a similar suggestion:
>     * The more attention heads a model has, the more likely it will be for at least one head to closely match the human pattern. At the same time, more attention heads will likely mean each individual head plays a less direct causal role in the behavior of the model on the prediction objective.
>     * We argue in the paper that models with a large number of attention heads are inconsistent with the way cognitive scientists think about working memory within cue-based retrieval theories. These theories emphasize a working memory system which is highly constrained in the number of retrieval operations that can take place during processing. If we are to view neural attention as a cognitive model of cue-based memory retrieval, multi-headed models do not transparently reflect individual retrieval processes. Regardless, if we find that attention measures in multi-headed transformer models fail to replicate the human patterns, then this provides an additional empirical argument for our model. If a measure from some multi-headed model does provide an equally good or better fit to the human data, then our argument on the grounds of cognitive plausibility would still hold.
> ---
> *Attention != faithful attention*
> * Thank you for the reference to Kobayashi et al. (2021). We agree with the reviewer that it will be useful to look at the weighted vector norms for each retrieved item. This would allow us to investigate whether context tokens which receive a large attention weight actually make up a large component of the resulting attention vector. We will include this analysis in the final paper.
> ---
> *“...humans can better judge the implausibility of verbs in the "double" condition, but relative attention does not...”*
> * Thank you for pointing out this discrepancy between the human and model behavior. Assuming you are referring to the difference in plausibility rating accuracies between condition (e) and conditions (b, d), we did not analyze these effects in the paper because these were not a focus of analysis in the original human experiment conducted by Laurinavichyute and von der Malsburg (2022). Because the primary focus of our paper was attraction effects, we only analyzed effects of attraction, matched for the number of violations. In the final paper, we will include a more detailed discussion of other discrepancies between model and human behavior, namely the discrepancy the reviewer points out, and the discrepancy between surprisal, attention, and human plausibility ratings in each of the no attractor conditions.
> ---
> *Releasing our model + data*
> * We regret that we were not able to release the model code and data in our initial submission. We will release all of the model and analysis code alongside the final version of the paper. For the full list of materials used in experiment 4.3, see Laurinavichyute and von der Malsburg (2022), who provide their data [here](https://onlinelibrary.wiley.com/action/downloadSupplement?doi=10.1111%2Fcogs.13086&file=cogs13086-sup-0001-SuppMat.pdf).

---

### Official Review · Reviewer_mtFz · 2023-08-06

**Soundness:** 5

**Excitement:**

4: Strong: This paper deepens the understanding of some phenomenon or lowers the barriers to an existing research direction.

**Paper Topic And Main Contributions:**

This paper evaluates a neural language model that integrates a recurrent structure with a very narrow (size=1) self-attention. The theoretical target is to model the role of cue-based memory retrieval during sentence comprehension. The model is evaluated in terms of whether it is sensitive to syntactic dependencies, and whether the attention distribution or next-word predictability match human sentence processing results from "agreement attraction" sentences that motivate relevant cue-based retrieval theories. The key contribution is that the single-unit attention mechanism offers a reasonable match to the data presented here, advancing earlier accounts that relied on non-human-like multi-headed attention mechanisms (e.g. GPT-2.)

**Questions For The Authors:**

This is a small point but given the large number of agreement attraction data that are available for re-analysis it is a shame that only a single data-set was compared to the model here. Even while I note this, I expect that a broader evaluation is already on the authors' agenda.



**Reasons To Accept:**

- Pursues an exciting link between cognitive science approaches to memory during language processing, and key developments in neural language modeling
- Likely of interest to psycholinguists, cognitive scientists, and neural network modelers comparing recurrent and transformer approaches for language.
- Clear theoretical motivation to address limitations of prior work
- Interesting evaluations in terms of language-modeling performance, sensitivity to dependencies, and match to human data
- Evaluation of adding supertagging training target along-side language modeling is a nice bonus

**Reasons To Reject:**

- Models are compared to a LSTM baseline, but given the incorporation of attention, would a (similarly-sized) Transformer baseline also be appropriate?
- It will be exciting to see the same method applied to other dependencies associated with cue-based retrieval (e.g. relative clause, ellipsis etc.) The focus here on subject-verb agreement alone certainly reflects the bulk of the literature but limits - at least to me -  the generalizability of the model presented here

**Reproducibility:**

4: Could mostly reproduce the results, but there may be some variation because of sample variance or minor variations in their interpretation of the protocol or method.

**Reviewer Confidence:**

4: Quite sure. I tried to check the important points carefully. It's unlikely, though conceivable, that I missed something that should affect my ratings.

---

> ### Author Rebuttal · Authors · 2023-08-28
>
> *Comparison to a Transformer Baseline*
> * Thank you for the great suggestion to compare results from the CBRNN model to a transformer model. We will include comparisons to attention and surprisal measures from a transformer model in the final paper. However, we believe that results from multi-headed models would come with a couple of caveats, which we include in a response to another reviewer with a similar suggestion:
>     * The more attention heads a model has, the more likely it will be for at least one head to closely match the human pattern. At the same time, more attention heads will likely mean each individual head plays a less direct causal role in the behavior of the model on the prediction objective.
>     * We argue in the paper that models with a large number of attention heads are inconsistent with the way cognitive scientists think about working memory within cue-based retrieval theories. These theories emphasize a working memory system which is highly constrained in the number of retrieval operations that can take place during processing. If we are to view neural attention as a cognitive model of cue-based memory retrieval, multi-headed models do not transparently reflect individual retrieval processes. Regardless, if we find that attention measures in multi-headed transformer models fail to replicate the human patterns, then this provides an additional empirical argument for our model. If a measure from some multi-headed model does provide an equally good or better fit to the human data, then our argument on the grounds of cognitive plausibility would still hold.
> ---
> *Investigating other dependency types/other datasets*
> * This is indeed an exciting avenue of future work that we are actively pursuing. Given space limitations, the focus of the paper is on motivating and presenting the model itself. We include one well studied dependency type (and two types of attraction) to demonstrate the model's effectiveness, and we will include additional dependency types in future a conference submission and a longer journal version of this work.

---

### Meta-Review · Area_Chair_D9FZ · 2023-09-12

**Recommendation:** 4

**Metareview:**

The paper under review introduces and evaluates language models with respect to human-like sentence processing, specifically focusing on cue-based memory retrieval and cognitive plausibility. The models are assessed in terms of their sensitivity to syntactic dependencies and their alignment with human sentence processing data, particularly in sentences involving agreement attraction. While the paper has notable strengths, such as clear theoretical motivation and interesting evaluations, there are concerns regarding the choice of baselines, generalizability, and the paper's organization.

Pros from the reviews:

- A reviewer highlights the paper's clear theoretical motivation to address limitations in prior work, making it valuable for researchers in both cognitive science and neural network modeling.

- A reviewer appreciates the incorporation of both expectation- and cue-based serial processing, which is a significant contribution, and suggests that the results support the authors' claims.

- A reviewer acknowledges the well-controlled experimental design and positive results, indicating that the proposed models can replicate human sentence processing behavior.

Cons from the reviews:

- Two reviewers express concerns about the absence of comparisons with baseline models, such as multi-head attention Transformers, which would help contextualize the contributions of the proposed models.

- One reviewer suggests that the paper's focus on subject-verb agreement may limit the generalizability of the model, and encourages exploring other dependencies associated with cue-based retrieval.

- Another reviewer criticizes the paper's writing, pointing out typos and the need for a more focused reorganization of the structure to improve readability.

Given the strong paper, good reviews and comprehensive rebuttals, the soundness of the paper is quite clear. Yet, the authors often explain that they will include additional results, explanation, data etc. in the final version of the paper. For an excellent contribution, these additional resources would be essential. Given the overall favourable reviews, I would like to see the paper at EMNLP.

---

### Decision · Program_Chairs · 2023-10-07

**Decision:**

Accept-Findings

**Comment:**

The paper under review introduces and evaluates language models with respect to human-like sentence processing, specifically focusing on cue-based memory retrieval and cognitive plausibility. The models are assessed in terms of their sensitivity to syntactic dependencies and their alignment with human sentence processing data, particularly in sentences involving agreement attraction. While the paper has notable strengths, such as clear theoretical motivation and interesting evaluations, there are concerns regarding the choice of baselines, generalizability, and the paper's organization.

Pros from the reviews:

- A reviewer highlights the paper's clear theoretical motivation to address limitations in prior work, making it valuable for researchers in both cognitive science and neural network modeling.

- A reviewer appreciates the incorporation of both expectation- and cue-based serial processing, which is a significant contribution, and suggests that the results support the authors' claims.

- A reviewer acknowledges the well-controlled experimental design and positive results, indicating that the proposed models can replicate human sentence processing behavior.

Cons from the reviews:

- Two reviewers express concerns about the absence of comparisons with baseline models, such as multi-head attention Transformers, which would help contextualize the contributions of the proposed models.

- One reviewer suggests that the paper's focus on subject-verb agreement may limit the generalizability of the model, and encourages exploring other dependencies associated with cue-based retrieval.

- Another reviewer criticizes the paper's writing, pointing out typos and the need for a more focused reorganization of the structure to improve readability.

Given the strong paper, good reviews and comprehensive rebuttals, the soundness of the paper is quite clear. Yet, the authors often explain that they will include additional results, explanation, data etc. in the final version of the paper. For an excellent contribution, these additional resources would be essential. Given the overall favourable reviews, I would like to see the paper at EMNLP.